# PCA of high dimensional random walks with comparison to neural network training

**Joseph M. Antognini**[*]
Whisper AI
joe.antognini@gmail.com

**Jascha Sohl-Dickstein**
Google Brain
jaschasd@google.com

## Abstract

One technique to visualize the training of neural networks is to perform PCA on the parameters over the course of training and to project to the subspace spanned by the first few PCA components. In this paper we compare this technique to the PCA of a high dimensional random walk. We compute the eigenvalues and eigenvectors of the covariance of the trajectory and prove that in the long trajectory and high dimensional limit most of the variance is in the first few PCA components, and that the projection of the trajectory onto any subspace spanned by PCA components is a Lissajous curve. We generalize these results to a random walk with momentum and to an Ornstein-Uhlenbeck processes (i.e., a random walk in a quadratic potential) and show that in high dimensions the walk is not mean reverting, but will instead be trapped at a fixed distance from the minimum. We finally analyze PCA projected training trajectories for: a linear model trained on CIFAR-10; a fully connected model trained on MNIST; and ResNet-50-v2 trained on Imagenet. In all cases, both the distribution of PCA eigenvalues and the projected trajectories resemble those of a random walk with drift.

## 1 Introduction

Deep neural networks (NNs) are extremely high dimensional objects. A popular deep NN for image recognition tasks, ResNet-50 (He et al., 2016), has ~25 million parameters for example, and it is common for language models to have more than one billion parameters (Jozefowicz et al., 2016). This overparameterization may be responsible for NN's impressive generalization performance (Novak et al., 2018). Simultaneously, the high dimensional nature of NNs makes them difficult to reason about.

Over the decades of NN research, the common lore about the geometry of the loss landscape of NNs has changed dramatically. In the early days of NN research it was believed that NNs were difficult to train because they tended to get stuck in suboptimal local minima. Later, Dauphin et al. (2014) argued that the true scourge of NN optimization was saddle points, not local minima. Choromanska et al. (2015) further used a spherical spin-glass model to conjecture that local minima of NNs are not much worse than global minima. Baity-Jesi et al. (2018) showed that in the typical case of an over-parameterized NN the dynamics of NN optimization are different from glassy systems, and claimed that the difficulties with NN optimization were instead due to vast plateaus where the gradient is very small. There has also been active debate as to whether the geometry of the loss landscape around minima can inform the NN's ability to generalize. Hochreiter & Schmidhuber (1997) and Keskar et al. (2017) have claimed that NNs that generalize better tend to find flatter minima, though Dinh et al. (2017) countered that due to the scale-free nature of NNs, there always exist sharp minima that generalize equally well.

---

[*]Work done as a Google AI Resident.

To help resolve these questions, we would ideally like to be able to visualize the loss landscapes of NNs, but this is a difficult, perhaps even futile, task because it involves embedding an extremely high dimensional space into very few dimensions — typically one or two. Goodfellow et al. (2015) introduced a visualization technique in which the loss is plotted along a straight line from the initial point to the final point of training (the "royal road"). The authors found that the loss often decreased monotonically along this path. They further considered the loss in the space from the residuals between the NN's trajectory to this royal road (note that while this is a two-dimensional manifold, it is not a linear subspace). Lorch (2016) and Lipton (2016) proposed another visualization technique in which principal component analysis (PCA) is performed on the NN trajectory and the trajectory is projected into the subspace spanned by the lowest PCA components. Lipton (2016) noted that most of the variance was in a small number of PCA components. Li et al. (2018) explored this technique in more depth by plotting 2-dimensional cross-sections of the loss landscape spanned by the first two PCA components.

In this paper we consider the theory behind this visualization technique. We show that PCA projections of random walks in flat space qualitatively have many of the same properties as projections of NN training trajectories. We then generalize these results to a random walk with momentum and a random walk in a quadratic potential, also known as an Ornstein-Uhlenbeck process (Uhlenbeck & Ornstein, 1930). This process is more similar to NN optimization since it consists of a deterministic component (the true gradient) plus a stochastic component. In fact, recent work has suggested that stochastic gradient descent (SGD) approximates a random walk in a quadratic potential (Ahn et al., 2012; Mandt et al., 2016; Smith & Le, 2018). Finally, we perform experiments on linear models and large NNs to show how closely they match this simplified model.

The approach we take to study the properties of the PCA of high dimensional random walks in flat space follows that of Moore et al. (2018), but we correct several errors in their argument, notably in the values of the matrix $\mathbf{S}^T\mathbf{S}$ and the trace of $(\mathbf{S}^T\mathbf{S})^{-1}$ in Eq. 10. We also fill in some critical omissions, particularly the connection between banded Toeplitz matrices and circulant matrices. We extend their contribution by proving that the trajectories of high dimensional random walk in PCA subspaces are Lissajous curves and generalizing to random walks with momentum and Ornstein-Uhlenbeck processes.

## 2 PCA of random walks in flat space

### 2.1 Preliminaries

Let us consider a random walk in $d$-dimensional space consisting of $n$ steps where every step is equal to the previous step plus a sample from an arbitrary probability distribution, $\mathcal{P}$, with zero mean and a finite covariance matrix.[2] For simplicity we shall assume that the covariance matrix has been normalized so that its trace is 1. This process can be written in the form

$$\mathbf{x}_t = \mathbf{x}_{t-1} + \boldsymbol{\xi}_t, \qquad \boldsymbol{\xi}_t \sim \mathcal{P}, \tag{1}$$

where $\mathbf{x}_t$ is a $d$-dimensional vector and $\mathbf{x}_0 = \mathbf{0}$. If we collect the $\mathbf{x}_t$s together in an $n \times d$ dimensional design matrix $\mathbf{X}$, we can then write this entire process in matrix form as

$$\mathbf{S}\mathbf{X} = \mathbf{R}, \tag{2}$$

where the matrix $\mathbf{S}$ is an $n \times n$ matrix consisting of 1 along the diagonal and -1 along the subdiagonal,

$$\mathbf{S} \equiv \begin{pmatrix} 1 & 0 & 0 & \cdots & 0 \\ -1 & 1 & 0 & \ddots & \vdots \\ 0 & -1 & 1 & \ddots & 0 \\ \vdots & \ddots & \ddots & \ddots & 0 \\ 0 & \cdots & 0 & -1 & 1 \end{pmatrix}, \tag{3}$$

and the matrix $\mathbf{R}$ is an $n \times d$ matrix where every column is a sample from $\mathcal{P}$. Thus $\mathbf{X} = \mathbf{S}^{-1}\mathbf{R}$.

To perform PCA, we need to compute the eigenvalues and eigenvectors of the covariance matrix $\hat{\mathbf{X}}^T\hat{\mathbf{X}}$, where $\hat{\mathbf{X}}$ is the matrix $\mathbf{X}$ with the mean of every dimension across all steps subtracted. $\hat{\mathbf{X}}$ can be found by applying the $n \times n$ centering matrix, $\mathbf{C}$:

$$\hat{\mathbf{X}} = \mathbf{C}\mathbf{X}, \quad \mathbf{C} \equiv \mathbf{I} - \frac{1}{n}\mathbf{1}\mathbf{1}^T. \tag{4}$$

We now note that the analysis is simplified considerably by instead finding the eigenvalues and eigenvectors of the matrix $\hat{\mathbf{X}}\hat{\mathbf{X}}^T$. The non-zero eigenvalues of $\hat{\mathbf{X}}^T\hat{\mathbf{X}}$ are the same as those of $\hat{\mathbf{X}}\hat{\mathbf{X}}^T$. The eigenvectors are similarly related by $\mathbf{v}_k = \mathbf{X}^T\mathbf{u}_k$, where $\mathbf{v}_k$ is a (non-normalized) eigenvector of $\hat{\mathbf{X}}^T\hat{\mathbf{X}}$, and $\mathbf{u}_k$ is the corresponding eigenvector of $\hat{\mathbf{X}}\hat{\mathbf{X}}^T$.

We therefore would like to find the eigenvalues and eigenvectors of the matrix

$$\hat{\mathbf{X}}\hat{\mathbf{X}}^T = \mathbf{C}\mathbf{S}^{-1}\mathbf{R}\mathbf{R}^T\mathbf{S}^{-T}\mathbf{C}, \tag{5}$$

where we note that $\mathbf{C}^T = \mathbf{C}$. Consider the middle term, $\mathbf{R}\mathbf{R}^T$. In the limit $d \gg n$ we will have $\mathbf{R}\mathbf{R}^T \to \mathbf{I}$ because the off diagonal terms will be $\mathbb{E}[\xi_i]^2 = 0$, whereas the diagonal terms will be $\mathbb{E}[\xi^2] = \sum_{i=0}^{d} \mathbb{V}[\xi_i] = 1$. (Recall that we have assumed that the covariance of the noise distribution is normalized; if the covariance is not normalized, this simply introduces an overall scale factor given by the trace of the covariance.) We therefore have the simplification

$$\hat{\mathbf{X}}\hat{\mathbf{X}}^T = \mathbf{C}\mathbf{S}^{-1}\mathbf{S}^{-T}\mathbf{C}. \tag{6}$$

## 2.2 Asymptotic convergence to circulant matrices

Let us consider the new middle term, $\mathbf{S}^{-1}\mathbf{S}^{-T} = (\mathbf{S}^T\mathbf{S})^{-1}$. The matrix $\mathbf{S}$ is a banded Toeplitz matrix. Gray et al. (2006) has shown that banded Toeplitz matrices asymptotically approach circulant matrices as the size of the matrix grows. In particular, Gray et al. (2006) showed that banded Toeplitz matrices have the same inverses, distribution of eigenvalues, and eigenvectors as their corresponding circulant matrices in this asymptotic limit (see especially theorem 4.1 and subsequent material from Gray et al. 2006). Zhu & Wakin (2017) has furthermore proved a stronger result that under some weak conditions all eigenvalues of a banded Toeplitz matrix are equal to all eigenvalues of a corresponding circulant matrix in the limit of large matrices. Thus in our case, if we consider the limit of a large number of steps, $\mathbf{S}$ asymptotically approaches a circulant matrix $\widetilde{\mathbf{S}}$ that is equal to $\mathbf{S}$ in every entry except the top right, where there appears a $-1$ instead of a $0$.[3]

With the limiting circulant behavior of $\mathbf{S}$ in mind, the problem simplifies considerably. We note that $\mathbf{C}$ is also a circulant matrix, and furthermore the product of two circulant matrices is circulant, the transpose of a circulant matrix is circulant, and the inverse of a circulant matrix is circulant. Thus the matrix $\hat{\mathbf{X}}\hat{\mathbf{X}}^T$ is asymptotically circulant as $n \to \infty$. Finding the eigenvectors is trivial because the eigenvectors of all circulant matrices are the Fourier modes. To find the eigenvalues we must explicitly consider the values of $\hat{\mathbf{X}}\hat{\mathbf{X}}^T$. The matrix $\mathbf{S}^T\mathbf{S}$ consists of a 2 along the diagonal, -1 along the subdiagonal and superdiagonal, and 0 elsewhere, with the exception of the bottom right corner where there appears a 1 instead of a 2.

While this matrix is not a banded Toeplitz, it is asymptotically equivalent to a banded Toeplitz matrix because it differs from a banded Toeplitz matrix by a finite amount in a single location (Böttcher et al., 2003). We now note that multiplication of the centering matrix does not change either the eigenvector or the eigenvalues of this matrix since all vectors with zero mean are eigenvectors of the centering matrix with eigenvalue 1, and all Fourier modes but the first have zero mean. Thus the eigenvalues of $\hat{\mathbf{X}}\hat{\mathbf{X}}^T$ can be determined by the inverse of the non-zero eigenvalues of $\mathbf{S}^T\mathbf{S}$, which is an asymptotic circulant matrix. The $k^{\text{th}}$ eigenvalue of a circulant matrix with entries $c_0, c_1, \dots$ in the first row is

$$\lambda_{\text{circ},k} = c_0 + c_{n-1}\omega_k + c_{n-2}\omega_k^2 + \dots + c_1\omega_k^{n-1}, \tag{7}$$

where $\omega_k$ is the $k^{\text{th}}$ root of unity. The imaginary parts of the roots of unity cancel out, leaving the $k^{\text{th}}$ eigenvalue of $\mathbf{S}^T\mathbf{S}$ to be

$$\lambda_{\mathbf{S}^T\mathbf{S},k} = 2\left[1 - \cos\left(\frac{\pi k}{n}\right)\right], \tag{8}$$

and the $k^{\text{th}}$ eigenvalue of $\hat{\mathbf{X}}\hat{\mathbf{X}}^T$ to be

$$\lambda_{\hat{\mathbf{X}}\hat{\mathbf{X}}^T,k} = \frac{1}{2}\left[1 - \cos\left(\frac{\pi k}{n}\right)\right]^{-1}. \tag{9}$$

The sum of the eigenvalues is given by the trace of $(\mathbf{S}^T\mathbf{S})^{-1} = \mathbf{S}^{-1}\mathbf{S}^{-T}$, and $\mathbf{S}^{-1}$ is given by a lower triangular matrix with ones everywhere on and below the diagonal. The trace of $(\mathbf{S}^T\mathbf{S})^{-1}$ is therefore given by

$$\text{Tr}\left(\mathbf{S}^{-1}\mathbf{S}^{-T}\right) = \frac{1}{2}n(n+1), \tag{10}$$

and so the explained variance ratio from the $k^{\text{th}}$ PCA component, $\rho_k$ in the limit $n \to \infty$ is

$$\rho_k \equiv \frac{\lambda_k}{\text{Tr}\left(\mathbf{S}^{-1}\mathbf{S}^{-T}\right)} = \frac{\frac{1}{2}\left[1 - \cos\left(\frac{\pi k}{n}\right)\right]^{-1}}{\frac{1}{2}n(n+1)}. \tag{11}$$

If we let $n \to \infty$ we can consider only the first term in a Taylor expansion of the cosine term. Requiring that $\sum_{k=1}^{\infty} \rho_k = 1$, the explained variance ratio is

$$\rho_k = \frac{6}{\pi^2 k^2}. \tag{12}$$

We test Eq. 12 empirically in Fig. 5 in the supplementary material.

We pause here to marvel that the explained variance ratio of a random walk in the limit of infinite dimensions is *highly* skewed towards the first few PCA components. Roughly 60% of the variance is explained by the first component, $\sim$80% by the first two components, $\sim$95% by the first 12 components, and $\sim$99% by the first 66 components.

## 2.3 Projection of the trajectory onto PCA components

Let us now turn to the trajectory of the random walk when projected onto the PCA components. The trajectory projected onto the $k^{\text{th}}$ PCA component is

$$\mathbf{X}_{\text{PCA},k} = \mathbf{X}\hat{\mathbf{v}}_k, \tag{13}$$

where $\hat{\mathbf{v}}_k$ is the normalized $\mathbf{v}_k$. We ignore the centering operation from here on because it changes neither the eigenvectors nor the eigenvalues. From above, we then have

$$\mathbf{X}_{\text{PCA},k} = \frac{1}{\|\mathbf{v}_k\|}\mathbf{X}\mathbf{v}_k = \frac{1}{\|\mathbf{v}_k\|}\mathbf{X}\mathbf{X}^T\mathbf{u}_k = \frac{\lambda_k}{\|\mathbf{v}_k\|}\mathbf{u}_k. \tag{14}$$

By the symmetry of the eigenvalue equations $\mathbf{X}\mathbf{X}^T\mathbf{u} = \lambda\mathbf{u}$ and $\mathbf{X}^T\mathbf{X}\mathbf{v} = \lambda\mathbf{v}$, it can be shown that

$$\|\mathbf{v}_k\| = \|\mathbf{X}^T\mathbf{u}_k\| = \sqrt{\lambda}. \tag{15}$$

Since $\mathbf{u}_k$ is simply the $k^{\text{th}}$ Fourier mode, we therefore have

$$\mathbf{X}_{\text{PCA},k} = \sqrt{\frac{2\lambda_k}{n}}\cos\left(\frac{\pi k t}{n}\right). \tag{16}$$

This implies that the random walk trajectory projected into the subspace spanned by two PCA components will be a Lissajous curve. In Fig. 1 we plot the trajectories of a high dimensional random walk projected to various PCA components and compare to the corresponding Lissajous curves. We perform 1000 steps of a random walk in 10,000 dimensions and find an excellent correspondence between the empirical and analytic trajectories. We additionally show the projection onto the first few PCA components over time in Fig. 6 in the supplementary material.

While our experiments thus far have used an isotropic Gaussian distribution for ease of computation, we emphasize that these results are completely general for *any* probability distribution with zero mean and a finite covariance matrix with rank much larger than the number of steps. We include the PCA projections and eigenvalue distributions of random walks using non-isotropic multivariate Gaussian distributions in Figs. 7 and 8 in the supplementary material.

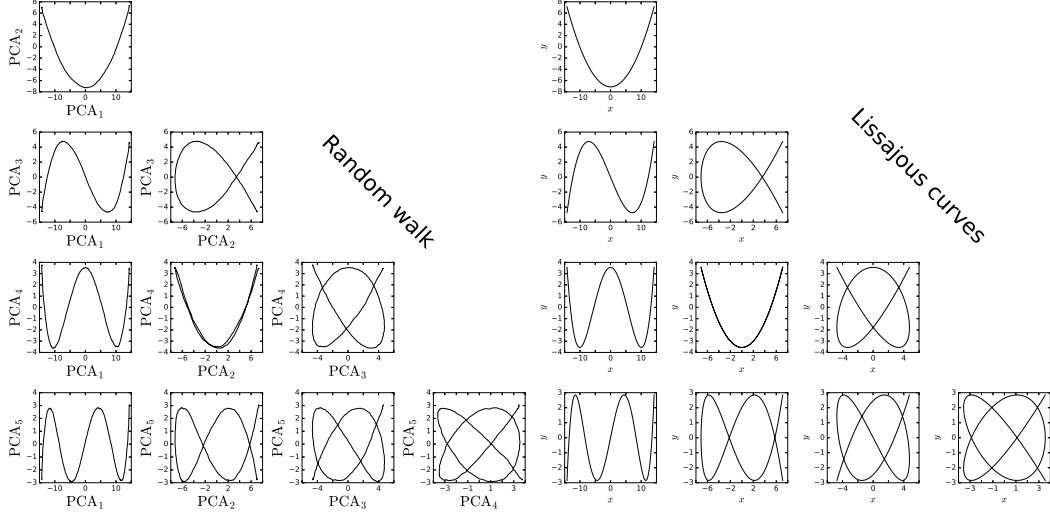

Figure 1: The PCA projections of the trajectories of high dimensional random walks are Lissajous curves. *Left tableau:* Projections of a 10,000-dimensional random walk onto various PCA components. *Right tableau:* Corresponding Lissajous curves from Eq. 16.

## 3 Generalizations

### 3.1 Random walk with momentum

It is a common practice to train neural networks using stochastic gradient descent with momentum. It is therefore interesting to examine the case of a random walk with momentum. In this case, the process is governed by the following set of updates:

$$
\begin{aligned}
\mathbf{v}_t &= \gamma \mathbf{v}_{t-1} + \boldsymbol{\xi}_t \tag{17} \\
\mathbf{x}_t &= \mathbf{x}_{t-1} + \mathbf{v}_t. \tag{18}
\end{aligned}
$$

It can be seen that this modifies Eq. 2 to instead read

$$
\mathbf{SX} = \mathbf{MR} \tag{19}
$$

where $\mathbf{M}$ is a lower triangular Toeplitz matrix with 1 on the diagonal and $\gamma^k$ on the $k^{\text{th}}$ subdiagonal. The analysis from Section 2 is unchanged, except that now instead of considering the matrix $\mathbf{S}^{-1}\mathbf{S}^{-T}$ we have the matrix $\mathbf{S}^{-1}\mathbf{M}\mathbf{M}^T\mathbf{S}^{-T}$. Although $\mathbf{M}$ is not a banded Toeplitz matrix, its terms decay exponentially to zero for terms very far from the main diagonal. It is therefore asymptotically circulant as well, and the eigenvectors remain Fourier modes. To find the eigenvalues consider the product $(\mathbf{S}^T\mathbf{M}^{-T}\mathbf{M}^{-1}\mathbf{S})^{-1}$, noting that $\mathbf{M}^{-1}$ is a matrix with 1s along the main diagonal and $-\gamma$s subdiagonal. With some tedious calculation it can be seen that the matrix $\mathbf{S}^T\mathbf{M}^{-T}\mathbf{M}^{-1}\mathbf{S}$ is given by

$$
(\mathbf{SM}^{-1}\mathbf{M}^{-T}\mathbf{S}^T)_{ij} = \begin{cases} 2 + 2\gamma + \gamma^2, & i = j \\ -(1+\gamma)^2, & i = j \pm 1 \\ \gamma, & i = j \pm 2 \\ 0, & \text{otherwise} \end{cases} \tag{20}
$$

with the exception that $\mathbf{S}_{nn} = 1$, and $\mathbf{S}_{n,n-1} = \mathbf{S}_{n-1,n} = -(1+\gamma)$. As before, this matrix is asymptotically circulant, so the eigenvalues of its inverse are

$$
\lambda_k = \frac{1}{2}\left[1 + \gamma + \gamma^2 - (1+\gamma)^2 \cos\left(\frac{\pi k}{n}\right) + \gamma \cos\left(\frac{2\pi k}{n}\right)\right]^{-1}. \tag{21}
$$

In the limit of $n \to \infty$, the distribution of eigenvalues is identical to that of a random walk in flat space, however for finite $n$, it has the effect of shifting the distribution towards the lower PCA components. We empirically test Eq. 21 in Fig. 9 in the supplementary material.

## 3.2 Discrete Ornstein-Uhlenbeck processes

A useful generalization of the above analysis of random walks in flat space is to consider random walks in a quadratic potential, also known as an AR(1) process or a discrete Ornstein-Uhlenbeck process. For simplicity we will assume that the potential has its minimum at the origin. Now every step consists of a stochastic component and a deterministic component which points toward the origin and is proportional in magnitude to the distance from the origin. In this case the update equation can be written

$$\mathbf{x}_t = (1 - \alpha)\mathbf{x}_{t-1} + \boldsymbol{\xi}_t, \tag{22}$$

where $\alpha$ measures the strength of the potential. In the limit $\alpha \to 0$ the potential disappears and we recover a random walk in flat space. In the limit $\alpha \to 1$ the potential becomes infinitely strong and we recover independent samples from a multivariate Gaussian distribution. For $1 < \alpha < 2$ the steps will oscillate across the origin. For $\alpha$ outside $[0, 2]$ the updates diverge exponentially.

### 3.2.1 Analysis of eigenvectors and eigenvalues

This analysis proceeds similarly to the analysis in Section 2 except that instead of $\mathbf{S}$ we now have the matrix $\mathbf{S}_{\text{OU}}$ which has 1s along the diagonal and $-(1 - \alpha)$ along the subdiagonal. $\mathbf{S}_{\text{OU}}$ remains a banded Toeplitz matrix and so the arguments from Sec. 2 that $\hat{\mathbf{X}}\hat{\mathbf{X}}^T$ is asymptotically circulant hold and its eigenvectors are remain Fourier modes. The eigenvalues will differ, however, because we now have that the components of $\mathbf{S}_{\text{OU}}^T \mathbf{S}_{\text{OU}}$ are given by

$$\left(\mathbf{S}_{\text{OU}}^T \mathbf{S}_{\text{OU}}\right)_{ij} = \begin{cases} 1 + (1-\alpha)^2, & i < n, i = j \\ -(1-\alpha), & i = j \pm 1 \\ 1, & i = j = n \\ 0, & \text{otherwise.} \end{cases} \tag{23}$$

From Eq. 7 we have that the $k^{\text{th}}$ eigenvalue of $\mathbf{S}_{\text{OU}}^T \mathbf{S}_{\text{OU}}$ is

$$\lambda_{\text{OU},k} = \left[1 + (1-\alpha)^2 - 2(1-\alpha)\cos\left(\frac{2\pi k}{n}\right)\right]^{-1} \simeq \left[\frac{4\pi^2 k^2 (1-\alpha)}{n^2} + \alpha^2\right]^{-1}. \tag{24}$$

We show in Fig. 2 a comparison between the eigenvalue distribution predicted from Eq. 24 and the observed distribution from a 3000 step Ornstein-Uhlenbeck process in 30,000 dimensions for several values of $\alpha$. There is generally a tight correspondence between the two. The exception is in the limit of $\alpha \to 1$, where there is a catch which we have hitherto neglected. While it is true that the *mean* eigenvalue of any eigenvector approaches the same constant, there is nevertheless going to be some distribution of eigenvalues for any finite walk. Because PCA *sorts* the eigenvalues, there will be a characteristic deviation from a flat distribution.

### 3.2.2 Critical distance and mixing time

While we might be tempted to take the limit $n \to \infty$ as we did in the case of a random walk in flat space, doing so would obscure interesting dynamics early in the walk. (A random walk in flat space is self-similar so we lose no information by taking this limit. This is no longer the case in an Ornstein-Uhlenbeck process because the parameter $\alpha$ sets a characteristic scale in the system.) In fact there will be two distinct phases of a high dimensional Ornstein-Uhlenbeck process initialized at the origin. First phase the process will behave as a random walk in flat space and the distance from the origin will increase proportionally to $\sqrt{n}$ and the variance of the $k^{\text{th}}$ PCA component will be proportional to $k^{-2}$. But once the distance from the origin reaches a critical value, the gradient toward the origin will become large enough to balance the tendency of the random walk to drift away from the origin.[4] At this point the trajectory will wander indefinitely around a sphere centered at the origin with radius given by this critical distance. Thus, while an Ornstein-Uhlenbeck process is mean-reverting in low dimensions, in the limit of infinite dimensions the Ornstein-Uhlenbeck process is no longer mean-reverting — an infinite dimensional Ornstein-Uhlenbeck process will never return to its mean.[5] This critical distance can be calculated by noting that each dimension is

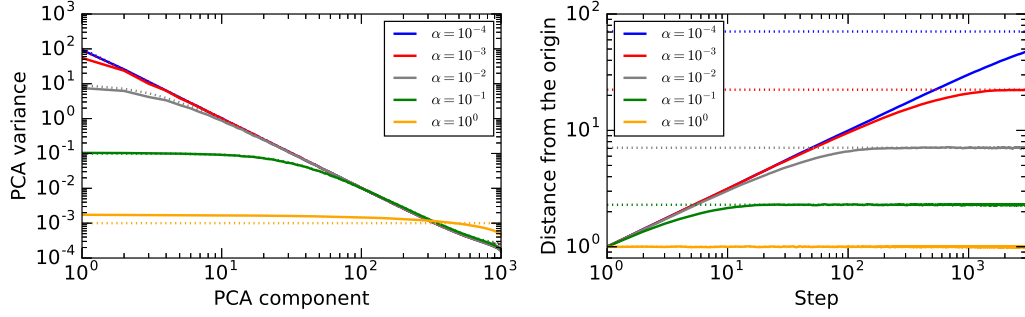

Figure 2: *Left panel:* The variance of the PCA components for several choices of $\alpha$. The empirical distribution is shown in solid and the predicted distribution with a dotted line. The predicted distribution generally matches the observed distribution closely, but there is a systematic deviation for $\alpha$ near 1. This is due to the fact that when the mean distribution is flat, there will nevertheless be a distribution around this mean when these eigenvalues are sampled from real data. Because PCA sorts these eigenvalues, this will always lead to a deviation from the flat distribution. *Right panel:* Distance from the origin for discrete Ornstein-Uhlenbeck processes with several choices of $\alpha$ (solid lines) with the predicted asymptote from Eq. 25 (dotted lines).

independent of every other and it is well known that the asymptotic distribution of an AR(1) process with Gaussian noise is Gaussian with a mean of zero and a standard deviation of $\sqrt{V/(1-(1-\alpha)^2)}$, where $V$ is the variance of the stochastic component of the process. In high dimensions the asymptotic distribution as $n \to \infty$ is simply a multidimensional isotropic Gaussian. Because we are assuming $V = 1/d$, the overwhelming majority of points sampled from this distribution will be in a narrow annulus at a distance

$$r_c = \frac{1}{\sqrt{\alpha(2-\alpha)}} \tag{25}$$

from the origin. Since the distance from the origin during the initial random walk phase grows as $\sqrt{n}$, the process will start to deviate from a random walk after $n_c \sim (\alpha(2-\alpha))^{-1}$ steps. We show in the right panel of Fig. 2 the distance from the origin over time for 3000 steps of Ornstein-Uhlenbeck processes in 30,000 dimensions with several different choices of $\alpha$. We compare to the prediction of Eq. 25 and find a good match.

### 3.2.3 Iterate averages converge slowly

We finally note that if the location of the minimum is unknown, then iterate (or Polyak) averaging can be used to provide a better estimate. But the number of steps must be much greater than $n_c$ before iterate averaging will improve the estimate. Only then will the location on the sphere be approximately orthogonal to its original location on the sphere and the variance on the estimate of the minimum will decrease as $1/\sqrt{n}$. We compute the mean of converged Ornstein-Uhlenbeck processes with various choices of $\alpha$ in Fig. 10 in the supplementary material.

### 3.2.4 Random walks in non-isotropic potential are dominated by low curvature directions

While our analysis has been focused on the special case of a quadratic potential with equal curvature in all dimensions, a more realistic quadratic potential will have a distribution of curvatures and the axes of the potential may not be aligned with the coordinate basis. Fortunately these complications do not change the overall picture much. For a general quadratic potential described by a positive semi-definite matrix $A$, we can decompose $A$ into its eigenvalues and eigenvectors. We then apply a coordinate transformation to align the parameter space with the eigenvectors of $A$. At this point we have a distribution of curvatures, each one given by an eigenvalue of $A$. However, because we are considering the limit of infinite dimensions, we can assume that there will be a large number of dimensions that fall in any bin $[\alpha_i, \alpha_i + d\alpha]$. Each of these bins can be treated as an independent high-dimensional Ornstein-Uhlenbeck process with curvature $\alpha_i$. After $n$ steps, PCA will then be dominated by dimensions for which $\alpha_i$ is small enough that $n \ll n_{c,i}$. Thus, even if relatively few

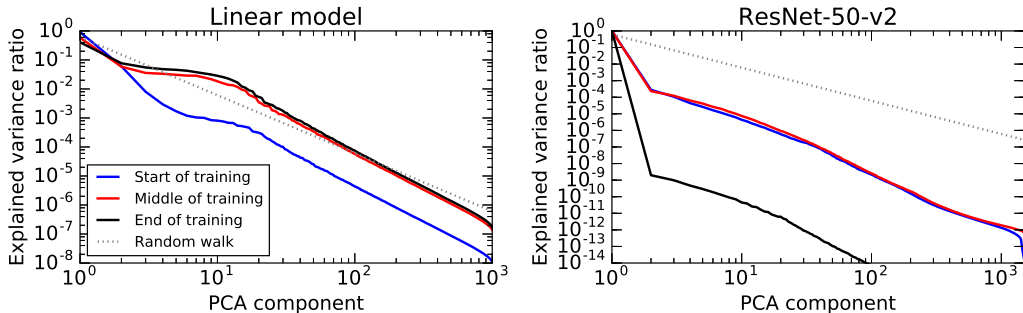

Figure 3: *Left panel:* The distribution of PCA variances at various points in training for a linear model trained on CIFAR-10. At the beginning of training the model's trajectory is more directed than a random walk, as exhibited by the steep distribution in the lower PCA components. By the middle of training this distribution has flattened (apart from the first PCA component) and more closely resembles that of an Ornstein-Uhlenbeck process. *Right panel:* The distribution of PCA variances of the parameters of ResNet-50-v2 at various points in training. The distribution of PCA variances generally matches that of a random walk with the exception of the first PCA component, which dominates the distribution, particularly at the end of training.

dimensions have small curvature they will come to dominate the PCA projected trajectory after enough steps.

## 4   Comparison to linear models and neural networks

While random walks and Ornstein-Uhlenbeck processes are analytically tractable, there are several important differences between these simple processes and optimization of even linear models. In particular, the statistics of the noise will depend on the location in parameter space and so will change over the course of training. Furthermore, there may be finite data or finite trajectory length effects.

To get a sense for the effect of these differences we now compare the distribution of the variances in the PCA components between two models and a random walk. For our first model we train a linear model without biases on CIFAR-10 using a learning rate of $10^{-5}$ for 10,000 steps. For our second model we train ResNet-50-v2 on Imagenet without batch normalization for 150,000 steps using SGD with momentum and linear learning rate decay. We collect the value of all parameters at every step for the first 1500 steps, the middle 1500 steps, and the last 1500 steps of training, along with collecting the parameters every 100 steps throughout the entirety of training. Further details of both models and the training procedures can be found in the supplementary material. While PCA is tractable on a linear model of CIFAR-10, ResNet-50-v2 has ∼25 million parameters and performing PCA directly on the parameters is infeasible, so we instead perform a random Gaussian projection into a subspace of 30,000 dimensions. We show in Fig. 3 the distribution of the PCA variances at the beginning, middle, and end of training for both models and compare to the distribution of variances from an infinite dimensional random walk. We show tableaux of the PCA projected trajectories from the middle of training for the linear model and ResNet-50-v2 in Fig. 4. Tableaux of the other training trajectories in various PCA subspaces are shown in the supplementary material, along with results from a small fully connected neural network trained on MNIST.

The distribution of eigenvalues of the linear model resembles an OU process, whereas the distribution of eigenvalues of ResNet-50-v2 resembles a random walk with a large drift term. The trajectories appear almost identical to those of random walks shown in Fig. 1, with the exception that there is more variance along the first PCA component than in the random walk case, particularly at the start and end points. This manifests itself in a small outward turn of the edges of the parabola in the PCA2 vs. PCA1 projection. This suggests that ResNet-50-v2 generally moves in a consistent direction over relatively long spans of training, similarly to an Ornstein-Uhlenbeck process initialized beyond $r_c$.

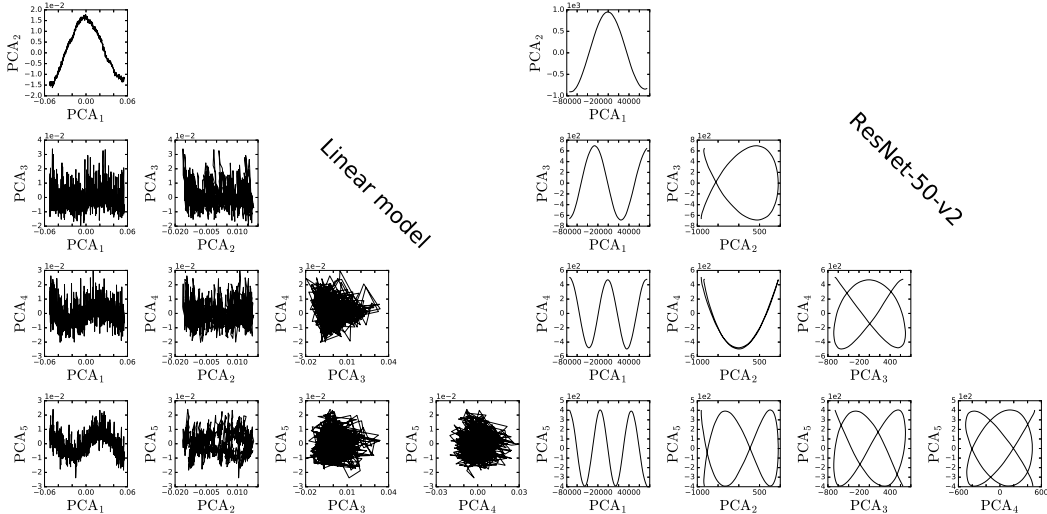

Figure 4: *Left tableau:* PCA projected trajectories from the middle of training a linear model on CIFAR-10. Training has largely converged at this point, producing an approximately Gaussian distribution in the higher PCA components. *Right tableau:* PCA projected trajectories from the middle of training ResNet-50-v2 on Imagenet. These trajectories strongly resemble those of a random walk. See Figs. 12 and 13 in the supplementary material for PCA projected trajectories at other phases of training.

## 5 Random walks with decaying step sizes

We finally note that the PCA projected trajectories of the linear model and ResNet-50-v2 over the entire course of training qualitatively resemble those of a high dimensional random walk with exponentially decaying step sizes. To show this we train a linear regression model $y = \mathbf{Wx}$, where $\mathbf{W}$ is a fixed, unknown vector of dimension 10,000. We sample $\mathbf{x}$ from a 10,000 dimensional isotropic Gaussian and calculate the loss

$$\mathcal{L} = \frac{1}{2}(y - y')^2, \tag{26}$$

where $y'$ is the correct output. We show in Fig. 15 that the step size decays exponentially. We fit the decay rate to this data and then perform a random walk in 10,000 dimensions but decay the variance of the stochastic term $\boldsymbol{\xi}_i$ by this rate. We compare in Fig. 16 of the supplementary material the PCA projected trajectories of the linear model trained on synthetic data to the decayed random walk. We note that these trajectories resemble the PCA trajectories over the entire course of training observed in Figs. 12 and 13 for the linear model trained on CIFAR-10 and ResNet-50-v2 trained on Imagenet.

## 6 Conclusions

We have derived the distribution of the variances of the PCA components of a random walk both with and without momentum in the limit of infinite dimensions, and proved that the PCA projections of the trajectory are Lissajous curves. We have argued that the PCA projected trajectory of a random walk in a general quadratic potential will be dominated by the dimensions with the smallest curvatures where they will appear similar to a random walk in flat space. Finally, we find that the PCA projections of the training trajectory of a layer in ResNet-50-v2 qualitatively resemble those of a high dimensional random walk despite the many differences between the optimization of a large NN and a high dimensional random walk.

**Acknowledgments**

The authors thank Matthew Hoffman, Martin Wattenberg, Jeffrey Pennington, Roy Frostig, and Niru Maheswaranathan for helpful discussions and comments on drafts of the manuscript.

## Footnotes

[2]The case of a constant non-zero mean corresponds to a random walk with a constant drift term. This is not an especially interesting extension from the perspective of PCA because in the limit of a large number of steps the first PCA component will simply pick out the direction of the drift (i.e., the mean), and the remaining PCA components will behave as a random walk without a drift term.

[3]We note in passing that $\widetilde{\mathbf{S}}$ is the exact representation of a *closed* random walk.

[4]Assuming we start close to the origin. If we start sufficiently far from the origin the trajectory will exponentially decay to this critical value.

[5]Specifically, since the limiting distribution is a $d$-dimensional Gaussian, the probability that the process will return to within $\epsilon$ of the origin is $P(d/2, \epsilon^2/2)$, where $P$ is the regularized gamma function. For small $\epsilon$ this decays exponentially with $d$.

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
