[Supplementary Material]

# 7 An overview of the theory of the asymptotic behavior of matrices

We here provide a brief review of several of the important concepts and theorems from the theory of the asymptotic convergence of matrix series. We start with statements of several standard definitions and theorems and then prove two corollaries that are relevant for our analysis in the main text. We refer the interested reader to the excellent review by Gray et al. (2006) for a more thorough introduction to the subject.

Recall that our ultimate goal is to demonstrate that two similar matrices have similar distributions of eigenvalues in the limit that the size of the matrices approaches infinity. In order to rigorously quantify the notion of similarity between matrices we turn to two kinds of matrix norm: the weak norm and the strong norm.

**Definition 1.** *Let $A$ be an $n \times n$ matrix, and let $\lambda_k$ be the eigenvalues of the matrix $A^* A$. Then the strong norm, $\|A\|$, is*

$$\|A\|^2 \equiv \max_k \lambda_k, \tag{27}$$

*and the weak norm, $|A|$, is*

$$|A|^2 \equiv \frac{1}{n} \sum_{i=0}^{n-1} \sum_{j=0}^{n-1} |a_{ij}|^2. \tag{28}$$

Note that because $\lambda_k \geq 0$, if $\|A\|$ is finite then $|A|$ must be finite as well. We now quote the definition of asymptotically equivalent sequences of matrices from Gray et al. (2006):

**Definition 2.** *"Two sequences of $n \times n$ matrices $A_n$ and $B_n$ are said to be asymptotically equivalent if*

*1. $A_n$ and $B_n$ are uniformly bounded in strong (and hence weak) norm:*

$$\|A_n\|, \|B_n\| \leq M < \infty, \quad n = 1, 2, \dots \tag{29}$$

*[where $M$ is some finite number] and*

*2. $A_n - B_n = D_n$ goes to zero in weak norm as $n \to \infty$:*

$$\lim_{n \to \infty} |A_n - B_n| = \lim_{n \to \infty} |D_n| = 0. \tag{30}$$

*Asymptotic equivalence of the sequences $A_n$ and $B_n$ will be abbreviated $A_n \sim B_n$."*

In the argument of the main text we take advantage of the fact that a sequence of matrices which differs from a sequence of banded Toeplitz matrices by a finite amount in a finite number of locations is asymptotically equivalent. Böttcher et al. (2003) and Rump (2006) have proved stronger results, but these are unfortunately not quite applicable to our case, so we prove the following theorem.

**Theorem 1.** *Consider two matrix sequences, $A_n$ and $T_n$, where $T_n$ is a banded Toeplitz matrix that is absolutely summable and bounded in strong norm by $M$. For any $n$, suppose that*

$$\max |a_{ij} - t_{ij}| \leq M' \tag{31}$$

*for some positive integer $M' < \infty$. Suppose furthermore that the number of entries $a_{ij} \neq t_{ij}$ is less than or equal to $M'$ and the discrepancies are all within a finite range $K$ of the diagonal. Then $A_n \sim T_n$.*

*Proof.* We must show that $A_n$ is bounded in the strong norm and that $A_n - T_n$ goes to zero in the weak norm. Let us first consider the strong norm of $A_n$, i.e., the maximum eigenvalue of $Z \equiv A_n^T A_n$. By the Gershgorin circle theorem, every eigenvalue of $Z$ lies within some disc centered on $z_{ii}$ with radius

$$R_{A,i} = \sum_{i \neq j} |z_{ij}|. \tag{32}$$

Similarly, every eigenvalue of $Y \equiv T_n^T T_n$ lies within some disc centered on $y_{ii}$ with radius

$$R_{T,i} = \sum_{i \neq j} |y_{ij}|. \tag{33}$$

Since $R_{T,i} < M$, $\max |z_{ij} - y_{ij}| \leq KM'$, and the maximum number of entries $z_{ij} \neq y_{ij}$ is less than or equal to $KM'$, we must have that $R_{A,i} \leq M + K^2 M'^2$. This means that the maximum eigenvalue of $A_n^T A_n$ is less than or equal to $M + K^2 M'^2$ and hence $A_n$ is bounded in strong norm.

Next consider the weak norm of $A_n$:

$$|A_n - T_n|^2 = \frac{1}{n} \sum_{i=0}^{n-1} \sum_{j=0}^{n-1} |a_n - t_n|^2 \tag{34}$$

$$\leq \frac{1}{n} M'^3. \tag{35}$$

And so

$$\lim_{n \to \infty} |A_n - T_n|^2 = 0. \tag{36}$$

Hence $A_n$ converges to $T_n$ in weak norm. $\qquad\square$

Crucial to the argument of the main text is the fact that banded Toeplitz matrices are asymptotically equivalent to circulant matrices. In particular, let $t_i$ be the number on the $i^{\text{th}}$ diagonal of a banded Toeplitz matrix, $T$, where $t_0$ is the main diagonal, $t_{-1}$ is the subdiagonal, $t_1$ is the superdiagonal, etc. For a Toeplitz matrix to be banded, we must have that $t_i = 0$ for $i > m$ and $i < -m$ for some positive integer, $m$. The equivalent circulant matrix, $C$, has entries $c_i = t_i$ for $-m < i < m$, and $c_{n-i+1} = t_i$ in order to satisfy the requirement that the matrix be circulant.

With these definitions in hand we can turn to a fundamental result in the theory of matrix sequences, that the eigenvalues of asymptotically equivalent sequences of matrices are asymptotically absolutely equally distributed. Again quoting from Gray et al. (2006):

**Theorem 2.** *"Let $A_n$ and $B_n$ be asymptotically equivalent sequences of Hermitian matrices with eigenvalues $\alpha_{n,k}$ and $\beta_{n,k}$ in nonincreasing order, respectively. ... [T]here exist finite numbers $m$ and $M$ such that*

$$m \leq \alpha_{n,k}, \beta_{n,k} \leq M, \quad n = 1, 2, \ldots, \quad k = 0, 1, \ldots, n-1. \tag{37}$$

*Let $F(x)$ be an arbitrary function continuous on $[m, M]$. Then*

$$\lim_{n \to \infty} \frac{1}{n} \sum_{k=0}^{n-1} |F(\alpha_{n,k}) - F(\beta_{n,k})| = 0. \tag{38}$$

*... [I]n this case the eigenvalues are said to be* asymptotically absolutely equally distributed.*"*

See theorem 2.6 of Gray et al. (2006) for a proof. By taking $F(x) = x$, this theorem implies that in the limit that $n \to \infty$ almost all eigenvalues of two asymptotically equivalent sequences of matrices are equal, except possibly for a set of measure zero. We can leverage this theorem to show that under certain conditions individual eigenvalues are equal between two asymptotically equivalent sequences of matrices. Zhu & Wakin (2017) have proved a similar, but stronger result, which is not quite applicable to our case because it concerns only banded Toeplitz matrices, whereas in our case we have a banded Toeplitz matrix with a localized impurity.

**Theorem 3.** *Let $A_n$ and $B_n$ be asymptotically equivalent sequences of Hermitian matrices with eigenvalues $\alpha_{n,k}$ and $\beta_{n,k}$ in nonincreasing order, respectively, and suppose there exist finite numbers $m$ and $M$ such that*

$$m \leq \alpha_{n,k}, \beta_{n,k} \leq M, \quad n = 1, 2, \ldots, \quad k = 0, 1, \ldots, n-1. \tag{39}$$

*If $|\alpha_{n,k} - \alpha_{n,k-1}| \geq \epsilon$ and $|\alpha_{n,k} - \alpha_{n,k+1}| \geq \epsilon$, for some $\epsilon > 0$ then $\alpha_{n,k} = \beta_{n,k}$.*

*Proof.* Let us take $F(x)$ to be the Gaussian function:

$$F_{\mu,\sigma}(x) \equiv \frac{1}{\sigma \sqrt{2\pi}} \exp\left(-\frac{1}{2\sigma^2}(x - \mu)^2\right). \tag{40}$$

Taking $\mu$ to be the eigenvalue $\alpha_{n,k}$, we have from Theorem 2,

$$\lim_{n \to \infty} \frac{1}{n} \sum_{j=0}^{n-1} |F_{\alpha_{n,k},\sigma}(\alpha_{n,j}) - F_{\alpha_{n,k},\sigma}(\beta_{n,j})| = 0. \tag{41}$$

Now, we have that $F_{\alpha_{n,k},\sigma}(\alpha_{n,k}) = 1/\sigma\sqrt{2\pi}$. Furthermore, the maximum value that this function can take for any other eigenvalue is

$$F_{\alpha_{n,k},\sigma}(\alpha_{n,k\pm1}) \leq \frac{1}{\sigma\sqrt{2\pi}} \exp\left(-\frac{\epsilon^2}{2\sigma^2}\right). \tag{42}$$

Hence Eq. 41 is bounded above by

$$\lim_{n \to \infty} \frac{n-1}{n} \left| \frac{1}{\sigma\sqrt{2\pi}} \exp\left(-\frac{\epsilon^2}{2\sigma^2}\right) \right| + \frac{1}{n} \left| \frac{1}{\sigma\sqrt{2\pi}} - \frac{1}{\sigma\sqrt{2\pi}} \exp\left(-\frac{(\alpha_{n,k} - \beta_{n,k})^2}{2\sigma^2}\right) \right|. \tag{43}$$

Now, the first term can be made as small as we like with a sufficiently small choice of $\sigma$. Similarly, if $\alpha_{n,k} \neq \beta_{n,k}$ we can make the second term be as close to $1/\sigma\sqrt{2\pi}$ as we like with a sufficiently small choice of $\sigma$. In order to make this expression tend to 0 in the limit $n \to \infty$ as required by Theorem 2, we must have that the difference between $\alpha_{n,k}$ and $\beta_{n,k}$ can be taken to be as small as we like for any sufficiently small choice of $\sigma$. This implies that in the limit of large $\sigma$ we have $\alpha_{n,k} = \beta_{n,k}$. □

# 8 Further empirical tests

## 8.1 High dimensional random walks

We test Eq. 12 by computing 1000 steps of a random walk in 10,000 dimensions and performing PCA on the trajectory. We show in Fig. 5 the empirical variance ratio for the various components compared to the prediction from Eq. 12 and find excellent agreement. The empirical variance ratio is slightly higher than the predicted variance ratio for the highest PCA components due to the fact that there are a finite number of dimensions in this experiment, so the contribution from all components greater than the number of steps taken must be redistributed among the other components, which leads to proportionally the largest increase in the largest PCA components.

We show in Fig. 6 the projection of the trajectory onto the first few PCA components. The projection onto the $k^{\text{th}}$ PCA component is a cosine of frequency $k/(2n)$ and amplitude given by Eq. 12.

## 8.2 Random walk with non-isotropic noise

To demonstrate that our results hold for non-isotropic noise distributions we perform a random walk where the noise is sampled from a multivariate Gaussian distribution with a random covariance matrix, $\Sigma$. Because sampling from a multivariate Gaussian with an arbitrary covariance matrix is difficult in high dimensions, we restrict the random walk to 1000 dimensions, keeping the number of steps 1000 as before. To construct the covariance matrix, we sample a $1000 \times 1000$ dimensional random matrix, $\mathbf{R}$, where each element is a sample from a normal distribution and then set $\Sigma = \mathbf{RR}^T$. Although $\Sigma$ will be approximately equal to the identity matrix, the distribution of eigenvalues will follow a fairly wide Marchenko-Pastur distribution because $\mathbf{R}$ is square. We show the distribution of explained variance ratios with the prediction from Eq. 12 in Fig. 7. There is a tight correspondence between the two up until the largest PCA components where finite dimension effects start to dominate. We also show in Fig. 8 PCA projected trajectories of this random walk along with a random walk where the random variates are sampled from a 1000-dimensional isotropic distribution for comparison to provide a sense for the amount of noise introduced by the relatively small number of dimensions. Although the small dimensionality introduces noise into the PCA projected trajectories, it is clear that the general shapes match the predicted Lissajous curves.

## 8.3 Random walk with momentum

We test Eq. 21 by computing 1000 steps of a random walk in 10,000 dimensions with various choices of the momentum parameter, $\gamma$. We show in Fig. 9 the observed distribution of PCA variances (not the explained variance ratio) along with the prediction from Eq. 21. There is an extremely tight

Figure 5: The fraction of the total variance of the different PCA components for a high dimensional random walk. The solid line is calculated from performing PCA on a 10,000 dimensional random walk of 1000 steps. The dashed line is calculated from the analytic prediction of Eq. 12. There is excellent agreement up until the very largest PCA components where finite size effects start to become non-negligible.

Figure 6: The projection of the trajectory of a high-dimensional random walk onto the first five PCA components forms cosines of increasing frequency and decreasing amplitude. The predicted trajectories are shown with dotted lines, but the difference between the predicted and observed trajectories is generally smaller than the width of the lines. The random walk in this figure consists of 1000 steps in 10,000 dimensions.

Figure 7: The distribution of explained variance ratios from the PCA of a random walk with noise sampled from a multivariate Gaussian with a non- isotropic covariance matrix. Despite the different noise distribution, the distribution of explained variance ratios closely matches the prediction from Eq. 12.

Figure 8: *Left tableau:* The PCA projected trajectory of a random walk with noise sampled from an isotropic Gaussian distribution in 1000 dimensions. *Right tableau:* The PCA projected trajectory of a random walk with noise sampled from a multivariate Gaussian distribution with a random covariance matrix in 1000 dimensions. Although the smaller number of dimensions introduces noise into the trajectory, it is clear that the trajectories are still Lissajous curves even when the random variates are sampled from a more complicated distribution.

Figure 9: The distribution of explained PCA variances for random walks with momentum where we vary the strength of the momentum parameter, $\gamma$. The distribution observed from a 1000 step random walk in 10,000 dimensions is shown in the solid lines. The prediction from Eq. 21 is shown in the dashed line.

Figure 10: The mean of all steps of converged Ornstein-Uhlenbeck processes of various lengths. The mean remains approximately constant until the total angle from the initial position on the sphere grows to $\sim \pi/2$, which requires $\sim \alpha^{-1}$ steps (dashed line).

correspondence between the two, except for the lowest PCA components for $\gamma = 0.999$. This is expected because the effective step size is set by $n/(1 - \gamma)$, and because $n = 1000$, the walk does not have sufficient time to settle into its stationary distribution of eigenvalues when $\gamma = 0.999$.

## 8.4 Iterate averaging of an Ornstein-Uhlenbeck process

We show in Fig. 10 the mean of all steps of Ornstein-Uhlenbeck processes which have converged to a random walk on a sphere of radius $r_c$. We show in the dashed line the predicted value of $n_c$, the number of steps required to reach $r_c$ (i.e., the crossing time of the sphere). The position on the sphere will close to its original location for $n \ll n_c$ so iterate averaging will not improve the estimate of the minimum. Only when $n \gg n_c$ will iterate averaging improve the estimate of the minimum since the correlation between new points wit the original location will be negligible.

# 9 Details of models and training

## 9.1 Linear regression on CIFAR-10

We train linear regression on CIFAR-10 for 10,000 steps using SGD and a batch size of 128 and a learning rate of $10^{-5}$. The model achieves a validation accuracy of 29.1%.

## 9.2 ResNet-50-v2 on Imagenet

We train ResNet-50-v2 on Imagenet for 150,000 steps using SGD with momentum and a batch size of 1024. We do not use batch normalization since this could confound our analysis of the training trajectory. We instead add bias terms to every convolutional layer. We decay the learning rate linearly with an initial learning rate of 0.0345769 to a final learning rate a factor of 10 lower by 141,553 steps, at which point we keep the learning rate constant. We set the momentum to 0.9842. The network achieves a validation accuracy of 71.46%.

# 10 Gallery of PCA projected trajectories

We present here tableaux of the PCA projections of various trajectories. We show in Fig. 11 four tableaux of the PCA projections of the trajectories of high-dimensional Ornstein-Uhlenbeck processes with different values of $\alpha$. For $\alpha = 10^{-4}$ the trajectories are almost identical to a high-dimensional random walk, as they should be since the process was sampled for only 1000 steps. Once we have $\alpha^{-1} = 1000$ the trajectories start to visibly deviate from those of a high-dimensional random walk. For larger $\alpha$ the deviations continue to grow until they become unrecognizable at $\alpha = 0.1$ because 1000 steps corresponds to many crossing times on the high dimensional sphere on which the process takes place.

In Fig. 12 we present tableaux of the PCA projections of the linear model trained on CIFAR-10. The trajectory of the entire training process somewhat resembles a high-dimensional random walk, though because the model makes larger updates at earlier steps than at later ones there are long tails on the PCA projected trajectories. The model's trajectory most closely resembles a high-dimensional random walk early in training, but towards the end the higher components become dominated by noise, implying that these components more closely resemble a converged Ornstein-Uhlenbeck process. This corresponds with the flattening of the distribution of eigenvalues in Fig. 3.

In Fig. 13 we present tableaux of the PCA projections of ResNet-50-v2 trained on Imagenet. Perhaps remarkably, these trajectories resemble a high-dimensional random walk much more closely than the linear model. However, as in the case of the linear model, the resemblance deteriorates later in training.

Finally, in Fig. 14 we present tableaux of the PCA projections of a two-layer fully connected neural network trained on MNIST. The neural network had 128 units in each hidden layer with ReLU activaitons and was trained for 10,000 steps with a learning rate of $10^{-4}$. As with the linear model we collect the parameters from the first 1000 steps, the middle 1000 steps, and the final 1000 steps, along with the parameters from every 10 steps throughout the entirety of training.

Figure 11: Tableaux of the PCA projections of the trajectories of high-dimensional Ornstein-Uhlenbeck processes with various values of $\alpha$. All processes were sampled for 1000 steps in 10,000 dimensions. *Upper left tableau:* $\alpha = 10^{-4}$. *Upper right tableau:* $\alpha = 10^{-3}$. *Lower left tableau:* $\alpha = 10^{-2}$. *Lower right tableau:* $\alpha = 10^{-1}$.

Figure 12: Tableaux of the trajectories of a linear model trained on CIFAR-10 in different PCA subspaces. *Upper left tableau:* PCA applied to every tenth step over all of training. *Upper right tableau:* PCA applied to the first 1000 steps of training. *Lower left tableau:* PCA applied to the middle 1000 steps of training. *Lower right tableau:* PCA applied to the last 1000 steps of training.

Figure 13: Tableaux of the parameter trajectories of ResNet-50-v2 trained on Imagenet in different PCA subspaces. The parameters were first projected into a random Gaussian subspace with 30,000 dimensions before PCA was applied. *Upper left tableau:* PCA applied to every hundredth step over all of training. *Upper right tableau:* PCA applied to the first 1500 steps of training. *Lower left tableau:* PCA applied to the middle 1500 steps of training. *Lower right tableau:* PCA applied to the last 1500 steps of training.

Figure 14: Tableaux of the parameter trajectories of a two-layer fully connected neural network trained on MNIST projected into different PCA subspaces. The parameters were first projected into a random Gaussian subspace with 10,000 dimensions before PCA was applied. *Upper left tableau:* PCA applied to every tenth step over all of training. *Upper right tableau:* PCA applied to the first 1000 steps of training. *Lower left tableau:* PCA applied to the middle 1000 steps of training. *Lower right tableau:* PCA applied to the last 1000 steps of training.

Figure 15: The change in the step size from training a linear model on synthetic Gaussian data. The step size decays exponentially with the best fit shown in the orange dashed line.

Figure 16: *Left tableau:* PCA projected trajectories of a linear regression model trained on synthetic Gaussian data. *Right tableau:* PCA projected trajectories of a 10,000 dimensional random walk where the variance of the stochastic component is decayed using the best fit found from the linear regression model trained on synthetic data. The trajectories in the two tableaux appear very similar.