[Reviews · NeurIPS 2018]

Reviewer 1



Motivated by the problem of visualizing the loss landscape of a deep neural networks during training, and the heuristic consisting of performing PCA on the set of parameters output by stochastic gradient descent, this paper considers a simplified model where the data comes from a simple random walk in Euclidean space instead of a NN. The authors attempt to justify this heuristic by asking what the projection of this walk on its first few principal components should look like. An asymptotic analysis is performed and the conclusion that most of the variance of the walk is captured by its first few components is reached. Furthermore, the projection of the walk in the first two components converges to a Lissajous curve, a parametric curve in the plane of the form t -> (cos(a t),cos(b t)). This reasoning is then extended to the case of a discrete Ornstein-Uhlenbeck process. Then the authors show that their findings are reasonably accurate compared to data coming from a NN on real-world datasets. The idea of performing PCA on the output of a NN for purposes of visualization is an interesting one, and this paper makes a first step towards understanding this proposal through the analysis of a very simple model. However, the analysis is conducted at an extremely precarious level of rigor. Limits from the discrete to the continuous are taken without care, and a number of approximations are made without making precise the meaning of the approximation (e.g. replacing Toeplitz by circular matrices, replacing discrete matrices by integral operators,…) while still writing equality symbols in the process. Also, the justifications related to the case of the OU process are more of qualitative discussion rather than a quantitative argument. Therefore the conclusions can only be trusted on a very qualitative level. To summarize, the idea is interesting but it seems that the authors have not spent a sufficient amount of effort into making their arguments quantitative and mathematically convincing.

Reviewer 2



The paper looks at the dynamics of training in DNNs through the lens of PCA in comparison with random walk. The work first looks at the behavior of first PCA components in random walk and compares it with the OU process. Then it looks at the DNNs and finds out that in the early phases of the training the behavior of weights are ballistic and later on they resemble more of an OU like process. The DNN part of the study is quite empirical but derivations are provided for the random walk. Recently, there have been several works on understanding the landscape of loss functions and dynamics on it. The flatness of the minima may imply diffusive behavior at the bottom of the landscape which is consistent with the findings in this paper as well (generalization connection: https://arxiv.org/abs/1609.04836, local geometry at the bottom (for instance discussion in 298-304 fits into this framework): https://arxiv.org/abs/1803.06969, connecting solutions at the bottom: https://arxiv.org/abs/1806.06977, diffusive (on non-glassy loss surface) dynamics at the bottom: https://arxiv.org/abs/1803.06969, PCA on NNs: https://arxiv.org/abs/1602.07320, etc...). It's pity that the current work doesn't discuss the current literature, but well, ML research these days can lack many things including scrutiny in experiments and I rather see a methodologically sound work than the one with missing core discussion. Speaking of the discussion, the paragraph between lines 22-30 has the correct conclusion, but the two papers cited there actually give different take-home messages. Dauphin et al is worried about the saddle points problem, and Choromanska et al is worried that the landscape will be mapped to a Hamiltonian of a spin glass (which is not true). The empirical results on finding the same loss values no matter what is interesting, but similar experiments can be found in several other works that do not necessarily claim that the landscape is glassy. Regarding the claims of the referred papers in lines 48-51, all three papers need to use a version of the functional central limit theorem which is very hard to justify in the case of DNN where the number of parameters is huge (for example, for reasons mentioned in lines 206-209!). This debate on whether SGD is Langevin type dynamics is a major debate and it is quite unclear that this is the case. It's another drawback of the paper that it just cites a few works for which that support their vague idea without any discussion in depth. Overall, I think the introduction and discussion part of the paper is quite poor and it can be misleading without precisely stating such pitfalls. The point of the above comments is not to include a massive discussion on every area, but to reduce the certainty in the language by picking a few papers that fit the general discussion. The paper has only 12 citations and 5 of them are quite non-essential as I have discussed above. At this stage of ML research, this is acceptable only if the work shows what they find. It is totally OK to show the results of their findings with minimal discussion, and the work will have its own value as a solid empirical work which is quite interesting and will have an impact in the community. But it is not OK to attempt to have a discussion with the missing sides of the story. A further note on experiments: It is appreciated that the experiments are carried on in large-scale models with real data. However, it would be great to see it in a small fully connected model, as well. Say on MNIST and 2 hidden layer fully connected model, how wide should the network be in order to see this behavior consistently? If it is claimed to work on ResNets, there must be a regime for which it works on FCs. Another advantage of such a model is that one won't need to pick a layer in the middle and can do PCA on the full weight vector. Also, for the ResNet, why only the sixth convolutional layer? Do you see this behaviour consistently in early layers and final layers as well? If not, why do you think that this is the case? Overall, I think the direction of the paper is great but it would have been much richer with further discussions, it would've been less misleading with fewer discussions. It would've been more complete with more experiments even if, say, the claim doesn't work for the last layers, for example. It is a good paper, but I think there is room for crucial improvement in the discussion section (either by expansion or removal), and further room for intermediate complexity experiments. (And of course, it would be great to have further information on how exactly the experiments are conducted for reproducibility...)

Reviewer 3



The paper gives a formal basis for the explanation of those techniques which operate pca to visualize the parameters of a neural network and how do they change during training. In the details, they show that PCA projections of random walks in flat space qualitatively have many of the same properties as projections of NN training trajectories. Substituting then the flat space with a quadratic potential, the Ornstein-Uhlenbeck process, which is more similar to NN optimization. This paper is a real pleasure to read. It tells things in a formal yet comprehensible fashion. Nonetheless, at the end, all of the formalization of the paper misses in my opinion what is the net value, apart for the formalization which is nice. Is it true that at some point I can do a sort of forecasting on how parameters will change during the training? Is there any way to decide, given the type of trajectory followed by the parameters, an early stop of the process of training? This can be put in the conclusions, which are very short and not very informative.